# Revealing the Therapeutic Targets and Mechanism of Ginsenoside Rg1 for Liver Damage Related to Anti-Oxidative Stress Using Proteomic Analysis

**DOI:** 10.3390/ijms231710045

**Published:** 2022-09-02

**Authors:** Jiying Hou, Ruoxiang Ma, Shisheng Zhu, Yaping Wang

**Affiliations:** 1Laboratory of Stem Cells and Tissue Engineering, Department of Histology and Embryology, Chongqing Medical University, Chongqing 400016, China; 2Faculty of Basic Medical Sciences, Chongqing Medical and Pharmaceutical College, Chongqing 401331, China

**Keywords:** ginsenoside Rg1, 4D-label-free proteomic analysis, oxidative stress, hepatic metabolism, D-galactose

## Abstract

Ginsenoside Rg1 is an important active substance isolated from the root of ginseng. In previous studies, Rg1 has shown excellent therapeutic effects in antioxidant, anti-inflammatory, and metabolic modulation. However, the therapeutic targets of Rg1 are still unknown. In this study, we investigated the therapeutic effects of Rg1 on oxidative stress-related liver damage. The oxidative stress damage model was achieved by intraperitoneal injection of D-galactose (D-gal) for 42 consecutive days in C57BL/6J mice. Rg1 treatment started on Day 16. Body weight, liver weight, degree of hepatic oxidative stress damage, serum lipid levels, and hepatic lipid and glucose metabolism were measured. Proteomics analysis was used to measure liver protein expression. The differential expression proteins were analyzed with bioinformatics. The results showed that Rg1 treatment attenuated liver damage from oxidative stress, reduced hepatic fat accumulation, promoted hepatic glycogen synthesis, and attenuated peripheral blood low-density lipoprotein (LDL), cholesterol (CHO), and triglycerides (TG) levels. Proteomic analysis suggested that Rg1 may regulate hepatocyte metabolism through ECM–Receptor, the PI3K-AKT pathway. The epidermal growth factor receptor (EGFR) and activator of transcription 1 (STAT1) may be the key protein. In conclusion, this study provides an experimental basis for further clarifying the specific mechanism of Rg1 in the treatment of oxidative stress damage-related liver disease.

## 1. Introduction

Free radicals are atoms or molecules with unpaired electrons, normally unstable and enormously reactive. Under ordinary conditions, the physique requires a certain level of reactive oxygen species(ROSs) to elevate its fundamental physiological features [1]. It is accountable for the expression of cellular functions, which include signal transduction pathways, defending in opposition to microbial invasion, and gene expression to promote growth or death. In contrast, when exposure to adverse stimuli or age-related oxidative stress accumulates in excess, giant quantities of ROS can result in cell death, leading to cell and tissue damage [2]. The liver performs a significant role in the body’s biochemical synthesis, with over 10,000 biochemical reactions taking place in the liver, such as gluconeogenesis, oxidative phosphorylation, and lipolysis [3]. These reactions involve carbohydrate, protein, and fat metabolism, exogenous metabolism, and many regulatory processes. The liver is the main organ of ROS attack. Oxidative stress is one of the pathological mechanisms that contribute to the development of many liver diseases [4], including chronic viral hepatitis, alcoholic liver disease, and non-alcoholic steatohepatitis.

D-gal is a typical modeling drug that mimics oxidative stress damage [5,6]. The pharmacological effect is that in the presence of galactose oxidase, excess D-gal may produce aldohexose and hydrogen peroxide and promote the production of oxygen-derived free radicals and superoxide anions. In liver research, D-gal has been used in many articles to model senescence and oxidative stress-related liver injury [7,8]. D-gal-treated animals showed reduced antioxidant enzyme activity, cell cycle arrest, chromatin damage, and other pathological changes associated with oxidative stress damage [9]. Therefore, our study used D-gal to trigger oxidative stress damage in the liver.

Ginseng is an ancient traditional medicinal herb that is widely used in Asia [10]. Rg1 is a natural pharmaceutical ingredient extracted from ginseng with anti-aging, antioxidant, and anti-inflammatory pharmacological effects [11,12,13,14]. In recent years, with the development of pharmacological studies on natural drugs, Rg1 has been identified as a potential acting component in ginseng [15]. Rg1 can improve insulin resistance and has shown good therapeutic potential in type 1 diabetes mellitus (T1DM), type 2 diabetes mellitus (T2DM) combined with fatty liver, diabetic cerebral infarction, and diabetic cardiomyopathy complications [16,17]. Our previous study also found that Rg1 alleviated aging liver damage, promoted the expression of antioxidant enzymes, and reduced steatosis in the liver [18,19]. However, past studies have mainly focused on observing alterations in relevant metabolic pathways following the effect of Rg1. The new scientific and technological tools represented by proteomics analysis can provide a broader view of the mechanisms underlying the development of natural medicines. In this study, we elucidated the key proteins and pathways of Rg1 that mitigate oxidative stress injury in the liver by 4D-Lable-free proteomic analysis combined with bioinformatics analysis. This study provides new insights and additional experimental data to understand the pharmacological mechanisms of Rg1 in greater depth.

## 2. Results

### 2.1. Validation of the D-Gal Damage Model and Observation of the Effect of Rg1 in Alleviating Oxidative Stress

D-gal is the classic drug for building models of senescence. D-gal induces widespread oxidative stress damage in vivo, manifested in particular by cell cycle arrest, reduces intracellular antioxidant capacity, and upregulates senescence-associated β-galactosidase (SA-β-gal) activity. The liver appearance of the D-gal group was yellow-tinged and greasy while the livers of the Rg1 + D-gal group were pink-red (Figure 1a). D-gal modeling resulted in weight loss (Figure 1d), a decrease in antioxidant enzymes superoxide dismutase (SOD) (Figure 1e), glutathione (GSH) (Figure 1h), catalase (CAT) (Figure 1f), and an increase in the oxidative stress products such as malondialdehyde (MDA) (Figure 1g). In contrast, Rg1 alleviated oxidative stress-induced damage and attenuated weight loss due to D-gal damage. Notably, there was a slight increase in liver weight following D-gal injury (Figure 1b), but there was no statistical difference in the ratio of liver weight (liver weight/body weight) between the treatment and Rg1 groups (Figure 1c). Senescence-associated β-galactosidase (SA-β-gal) staining is the gold standard method for detecting senescent cells. The staining of specimens revealed a significant upregulation of SA-β-gal activity in the liver of mice in the D-gal group, suggesting an increase of senescent cells in the liver of D-gal group mice. In contrast, Rg1 attenuated the increase of senescent cells in the liver caused by D-gal (Figure 1i,j). Concurrently, the early markers of cellular senescence such as P16 and P21 were significantly elevated after D-gal treatment. The nuclear damage marker phospho-histone H2A.X (γ-H2A.X) was also significantly increased in the D-gal group. The expression of these proteins was significantly reduced by Rg1 treatment (Figure 1k,l). The above experiments showed that D-gal treatment can induce oxidative stress damage in liver cells which leads to the senescence of hepatocytes, while Rg1 can alleviate oxidative stress damage.

### 2.2. Rg1 Alleviated Liver Function Damage in D-Gal-Treated Mice

Oxidative stress-induced senescence plays an important role in the pathogenesis of several hepatic metabolic diseases, including nonalcoholic fatty liver disease (NAFLD). To observe whether Rg1 could alleviate oxidative dress-related liver dysfunction, the peripheral serum of mice in both D-gal and D-gal + Rg1 groups was measured through biochemical testing. Alanine transaminase (ALT) (Figure 2a) and aspartate aminotransferase (AST) (Figure 2b) are important biochemical enzymes that reflect liver function. Rg1 was able to reduce the D-gal-induced increase of ALT and AST. In addition, tests of peripheral serum lipid levels showed that Rg1 was able to reduce the elevation of CHO, TG, and LDL caused by D-gal. The levels of high-density lipoprotein (HDL) between the two groups were not statistically different (Figure 2c).

Morphological observations were made on the liver of both groups of mice. Hematoxylin and eosin (HE) staining (Figure 2d) showed more vacuoles in hepatocytes in the D-gal group and Rg1 treatment significantly reduced the number of vacuoles. Oil Red O staining (Figure 2f) is the classic test for the detection of lipogenesis, along with the use of Periodic acid–Schiff (PAS) staining (Figure 2g) for the detection of intracellular glycogen in hepatocytes. The results showed that D-gal induced intrahepatocellular fat accumulation (Figure 2i) and decreased hepatic glycogen synthesis (Figure 2j). In contrast, Rg1 treatment reduced the extent of fat accumulation, while hepatic glycogen synthesis increased. Masson staining (Figure 2e) was able to detect the extent of fibrous tissue hyperplasia in the liver, with significant fibrous hyperplasia in the accessory hepatic hilar region after D-gal treatment, while Rg1 was able to reduce the extent of fibrous hyperplasia (Figure 2h). Quantification of the stained smears showed a statistically significant difference in the comparison between the two groups.

The above experimental results indicated that Rg1 could alleviate the structural and functional damage to the liver caused by D-gal, reduce fat accumulation, promote hepatic glycogen synthesis, and reduce fibrosis, thus improving the disorder of lipid metabolism caused by oxidative stress-related liver damage.

### 2.3. The Results of 4D-Label-Free Proteomic Analysis of Liver Tissue

4D-Label-free proteomic analyses were performed on liver tissue from the D-gal and D-gal + Rg1 groups. In total, 3614 proteins were found to be expressed. The protein with a coefficient of variation of 1.5 (fold change of ≥1.5 or ≤0.67) and *p* ≤ 0.05 was defined as a differential express protein (DEPs). Rg1 + D-gal/D-gal, with a total of 306 differential proteins identified. Of these, 224 were up-regulated and 82 were down-regulated. Details of the differential proteins are listed in Appendix A. The raw data are in Appendix A.

To study the changes in DEPs in the livers of the two groups of mice, it is important to understand the function, cellular localization, and biological processes of each protein. Therefore, we used the functional annotations of the DAVID database (2021 version) for gene ontology (GO) functional enrichment analysis. The significant difference was identified as a false discovery rate (FDR) ≤ 0.05. GO functional analysis includes three categories: biological processes (BP), molecular functions (MF), and cellular components (CC). The top 10 sections of each analysis were listed based on FDR for presentation. In BP (Figure 3a), cell adhesion, cell-matrix adhesion, integrin-mediated signaling pathway, and the glycolytic process have changed significantly. In CC (Figure 3b), extracellular space, the myosin complex, the extracellular matrix, and the basement membrane showed a significant difference. In MF (Figure 3c), actin filament binding, the extracellular matrix structural constituent, actin binding, integrin binding, and calcium ion binding were the main functional category. The results of the GO analysis suggested that altered cell-matrix composition, altered membrane receptors, and enhanced cellular responses to the external may be the main biological processes of the effect of Rg1 in regulating the resistance to oxidative stress in the liver.

Next, we enriched the pathways of these differentially expressed proteins by Kyoto encyclopedia of genes and genomes (KEGG) enrichment analysis. The pathway was listed based on FDR from low to high for presentation (Figure 4). Notably, ECM–receptor interaction and focal adhesion were the most enriched pathways. In addition, among the top 10 KEGG enrichments, the pathway significantly associated with differences in glycolipid metabolism was the PI3K-AKT signaling pathway. The KEGG enrichment results were consistent with the GO results, suggesting that the role of Rg1 may be related to ECM–mediated activation of integrins and downstream alterations in the PI3K-AKT pathway.

### 2.4. Bioinformatics Analysis of Key Proteins

To further understand the interactions between DEPs, the String database was queried to identify interactions between the target proteins and other proteins that act directly on them. Analysis was performed with Cytoscape software. Protein-protein interaction (PPI) analysis using the Degree algorithm in the cytoHubba plugin yielded a relatively focused network (Figure 5b). The resulting network included 50 proteins, which were used as key proteins for further analysis. The colors of the proteins in the network represent the degree score. The redder color means a higher score. Proteins with a Degree score of 30–35 are in the inner circle, those with a score of 20–30 are in the middle circle, and those with a score of 5–20 are in the outer circle. The relative expression abundance between key proteins was shown (Figure 5a).

The GO analysis of key proteins showed: In CC (Figure 6a), the cell junction, the external side of the plasma membrane, and the focal adhesion showed a significant difference. In BP (Figure 6a), cell adhesion, cell-cell adhesion, and the integrin-mediated signaling pathway have changed significantly. In MF (Figure 6c), actin filament binding, actin binding, and integrin binding were the main functional categories. The results of the GO analysis suggested that the integrin family and its downstream pathways and proteins may play an important role in the effect of Rg1.

The KEGG enrichment analysis showed (Figure 7) that the ECM–receptor interaction and focal adhesion were still the most enriched pathways. The PI3K-AKT signaling pathway may also play a significant role as an activated downstream pathway of the integrin family. The KEGG enrichment results were consistent with the GO results, suggesting that the integrin family and downstream PI3K-AKT pathway may be related to the effect of Rg1.

### 2.5. Parallel Reaction Monitoring (PRM) Validates Proteomic Results

The results of GO, KEGG, and PPI suggested that integrin family proteins and the downstream PI3K-AKT pathway may play an important role in Rg1-mediated anti-oxidative stress damage in the liver of mice. We sought to validate the link between Rg1 antioxidant action by relative quantitative analysis of PRM. From the key proteins, we selected eight proteins (Egfr, Stat1, Itgav, Col1a1, Col4a1, Lama5, Col6a1 and Fn1) for quantitative analysis by liquid chromatography-tandem mass spectrometry (LC-PRM/MS). Of these eight proteins, five were associated with the PI3K-AKT pathway, and two with ECM–receptor-related cellular responses were selected.

Seventeen peptides of eight target proteins from two groups were quantified by LC-PRM/MS and analyzed by Skyline (https://skyline.ms/) (accessed on 16 June 2022). Quantitative information on the target peptide was found, as detailed in Appendix A. The quantitative information is normalized by isotope relabeling of the peptides, followed by quantitative analysis of the target peptides and target proteins. We performed PRM assays to confirm the mass spectrometry results in the D-gal group and D-gal + Rg1 group. The expression levels of these eight proteins showed a consistent trend with the mass spectrometry results (Figure 8).

## 3. Discussion

In this study, 4D-label-free based proteomic analysis and mass spectrometry techniques were applied in combination with molecular biology experiments to reveal the therapeutic mechanism of Rg1 in the treatment of D-gal-induced oxidative stress damage in the liver of mice. D-gal can react with proteins in vitro and in vivo to form advanced glycosylation end products (AGE) [20] and lead to the accumulation of ROS [21], which is thought to be closely related to senescence. Senescence is associated with a progressive decline in antioxidant enzymes. SOD catalyzes the conversion of superoxide anions to hydrogen peroxide and GSH breaks down hydrogen peroxide into oxygen and water in the cytoplasm. In the present study, antioxidant protection was disturbed in mice after 6 weeks of D-gal treatment. In contrast, levels of SOD, GSH, and CAT were restored in the Rg1-treated group, indicating that Rg1 was able to improve senescence-related oxidative stress in the liver. At the same time, the tests of liver function showed that D-gal-induced oxidative stress damage leads to hepatic steatosis as well as reduced liver function, which is consistent with the pathological changes in liver disease induced by aging in real life. In contrast, Rg1 treatment may provide some relief from hepatic steatosis and reduced liver function due to oxidative stress injury.

Rg1 is an important active substance extracted from ginseng. Previous studies have shown activity in anti-oxidative stress and anti-inflammatory responses [18,19]. In studies on the hepatoprotective active substances of ginseng, Rg1 has been identified as an important active substance that can alleviate insulin resistance and reduce the level of oxidative stress damage in the liver [22]. It has shown significant therapeutic activity in the treatment of oxidative stress-related liver diseases such as NALFD [18].

The hepatocyte function is regulated by many factors. In this study, a combination of proteomic analysis and bioinformatics analysis of key proteins revealed that the expression of multiple integrin proteins, as well as EGFR, was increased in the Rg1 treatment group. EGFR is a cell surface protein that can bind to a variety of ligands. Within the cell, EGFR can form homo- or heterodimers with other EGFR family members upon binding to the corresponding ligands. Upon activation of the intrinsic kinase structural domain, the main downstream signaling pathways activated are the Ras-Raf-MEK-ERK1/2 and signal transducer and activator of transcription (STAT) 3 and 5 pathways controlling the proliferation and differentiation and phosphatidylinositol-3-kinase (PI3K)-Akt -the mechanistic target of the survival-controlling rapamycin (mTOR) pathway [23,24,25]. At the same time, EGFR and integrin proteins, especially α5β1, play an important role in the hydration of hepatocytes as mechanical stress sensors [26,27]. In the normal organism, insulin triggers PI3 kinase-driven phosphorylation and activation of Na^+^/K^+^/2Cl^−^ cotransporter (NKCC1), which triggers cell swelling and activates integrin-dependent osmotic sensing and signaling [28,29]. This leads to the inhibition of protein hydrolysis and EGFR activation [30]. In contrast, pathological factors such as oxidative stress and hypernatremia can resist this action of insulin and thus become pro-diabetic [31]. Integrin proteins such as EGFR and alpha v integrins (Itgav) were significantly elevated in the results of protein interaction analysis. At the same time, various proteins identified by the proteome such as Itga9, Itgb3, Lamb2, Itgav, Col1a2, Col1a1, Itga3, Col4a1, Lamc1, Lama5, VWF, Col6a1, Fn1, and Itga2b are involved in the regulation of the PI3K-AKT pathway. This is an interesting finding, as there have been many studies on the regulation of liver metabolism by Rg1 in the PI3K-AKT pathway [18]. However, in our experiments, through proteomic and bioinformatics analysis, the results have led to new insights into the therapeutic pathways of Rg1. As the highly significant changes in the integrin system, ECM–receptor pathways, and cytoskeletal proteins in hepatocytes following Rg1 treatment suggested that Rg1 may more likely alter biological events in hepatocytes by promoting these pathways and proteins.

On the other hand, the analysis of other key proteins showed similar results. STAT1 is also one of the downstream proteins bound by EGFR. STAT1 is an important negative regulator of liver fibrosis [32,33]. STAT1 activation enhances STAT3-dependent proliferation in hepatocytes [34]. The upregulation of STAT1 may be related to the effect of Rg1 in reducing liver fibrosis.

However, the EGFR signaling pathway is a double-edged sword. The EGFR signaling system has been shown in many studies to play a central role in the repair and regenerative response to liver injury and inflammation, including inhibition of intrahepatic lipid accumulation [35]. Its role is in promoting the recovery of liver function in the presence of certain inflammatory damage. However, the EGFR signaling system also plays an important role in the development of liver cancer [36]. Studies have shown that the use of EGFR inhibitors can limit the expansion of hepatocellular carcinoma [37]. Therefore, the relationship between the degree of EGFR activation and Rg1 dose is a question that needs to be explored. However, the relationship between the concentration of Rg1 and the degree of oxidative stress damage was not discussed in this study, which reflects the limitations of this study and needs to be discussed in the next experiments.

Ginseng has a long history as a medicinal herb, and its anti-inflammatory and antioxidant effects have been confirmed by many studies. The therapeutic effects of Rg1, an important active component of ginsenosides, in terms of metabolism have also been demonstrated in many studies. However, the mechanism is still being elucidated. This study combines in vivo models with proteomic analysis and bioinformatics approaches to provide new insights into the pathway of action of Rg1.

## 4. Materials and Methods

### 4.1. Animal Experiments

C57BL/6J was selected as the experimental animal based on the completion of whole genome sequencing and the stable genetic background [38]. The C57BL/6J mice (6–8 weeks, equal numbers of male and female mice) used for experiments were purchased from and housed at the Animal Experimentation Centre of Chongqing Medical University. Previous experiments have ruled out a possible effect of Rg1 on the motile cycle of female mice [39]. All procedures were approved by the Animal Ethics Committee of Chongqing Medical University.

Rg1 was purchased from Biological Reagents (MCE, Cat. No. HY-N0045 or Jilin Hongjiu Biotechnology Co., Ltd., Jilin, China). The dose used in the experiment was 20 mg/kg day. With regard to the safety of Rg1, we have tested it in several organ systems throughout the body in past experiments [39,40,41,42,43]. The results showed no cytotoxic or other organ-damaging effects of this dose of Rg1. In the liver, experiments based on a negative control group have been published in advance [18,19]. Therefore, the mice were randomly divided into two groups (the D-gal group and the Rg1 + D-gal group).

D-gal was purchased from Biological Reagents (MCE, Cat. No. HY-N0210). D-gal is used in doses of 120 mg/kg. day. D-gal was injected intraperitoneally for 42 days. For the Rg1 + D-gal group, intraperitoneal injection of Rg1 was added 16 days after intraperitoneal injection of D-gal (>12 h between injections of the two drugs), respectively, for a total of 27 days. Mice in the modeling and dosing group were euthanized at the end of modeling using CO_2_ and deep anesthesia with pentobarbital (50 mg/kg) [44].

### 4.2. Antioxidant Enzymes and Redox Products

#### 4.2.1. Protein Concentration Measurement by BCA Protein Assay Reagent

Blood was cleared from the liver using a perfusion of 0.9% NaCl and 0.16 mg/mL sodium heparin. The liver was mixed at a ratio of 10 mg tissue to 100 μL lysate, lysed on ice for 30 min, and homogenized by ultrasound every 10 min. Then the samples were centrifuged at 10,000× *g* for 10 min at 4 °C. The supernatant was collected. The BCA reagent (P0012, Beyotime Co., Shanghai, China) was used to determine the protein concentration of the supernatant.

#### 4.2.2. Detection of SOD, CAT, and MDA

Samples for SOD, CAT, and MDA were separated and assayed the protein concentration with a BCA protein assay reagent. The kit to detect oxidative stress levels was purchased from Beyotime Biotechnology (Shanghai, China). The CAT (S0051): standards and samples were incubated in 96 well plates at 25 °C for 20 min. Afterward, the optical density value was measured at 520 nm. The final CAT content was calculated according to the instructions. The SOD (S0101S): The protein concentration was adjusted to 20–100 μg/sample. Then incubating in a 96 well plate at 37 °C for 30 min and detecting optical density value at 450 nm. The SOD content was calculated according to the instructions. The MDA(S0131S): The sample was mixed with the working solution and heated at 100 °C for 15 min. Then the mixture was centrifuged at 1000× *g* for 10 min at room temperature and 200 μL of the working solution was taken in a 96 well plate to determine the optical density value at 532 nm. The MDA content was calculated according to the instructions.

#### 4.2.3. Detection of GSH

The GSH (S0053): Tissue is frozen in liquid nitrogen and quickly ground to a powder, then centrifuged at 10,000× *g* for 10 min at 4 °C after a thorough shaking reaction with protein removal reagent. Fifty microliters of 0.5 mg/mL NADPH solution were added. Measuring the optical density at 412 nm every 5 min for a total of 25 min at 25 °C. Calculating the GSH concentration according to the equation in the instructions.

### 4.3. Senescence-Associated β-Galactosidase (SA-β-Gal) Staining and Oil Red O Staining

Fresh tissue was embedded with a frozen section embedding agent and sliced at −20 °C. SA-β-gal staining was performed using the senescence-associated β-galactosidase staining kit (C0602, Beyotime Co., Shanghai, China). Oil Red O staining was performed using Oil Red O (OILR-10001, Cyagen Biosciences, Guangzhou, China). After staining, the nuclei were re-stained and then sealed for observation.

### 4.4. Immunoblot Assay

Tissue proteins were extracted after ultrasonic lysis and the protein concentration was measured by BCA reagent (P0012, Beyotime Co., Shanghai, China). Electrophoresis was carried out using SDS-page gels (PG110, Epizyme Biomedical Technology Co., Ltd., Shanghai, China), after which the proteins were electrotransferred to PVDF membranes (WJ001, Epizyme Biomedical Technology Co., Ltd., Shanghai, China). The closure was performed using 5% nonfat milk. Primary antibodies were used with P16 (16717-1-AP, 1:1000, ProteinTech, Wuhan, China), P21 (28248-1-AP, 1:1000, ProteinTech, Wuhan, China), γ-H2A.X(#9718S, 1:1000, Cell Signaling Technology Co., Ltd., Shanghai, China), and β-actin (81115-1-RR, 1:10,000, ProteinTech, Wuhan, China) and closed overnight at 4 °C. This was followed by incubation with the secondary antibody for 1 h at room temperature. The immunoblots were analyzed with the ChemiDoc Touch Imaging System (Bio-Rad Laboratories, Inc., Hercules, CA, USA). Measurement of protein expression of β-actin was used as an internal reference. Relative densitometry analysis was carried out with Image J software (National Institutes of Health, Bethesda, MD, USA) (Version 1.52).

### 4.5. Paraffin Section

Hematoxylin–eosin staining (HE), Masson staining, and PAS staining was performed using paraffin sections. The gradient-dehydrated tissue was paraffin-embedded and sectioned at 5 mm. HE staining was performed using hematoxylin-eosin stain (C0105S, Beyotime Co., Shanghai, China). Masson was stained with potassium dichromate overnight, followed by ferric hematoxylin, lixin acid magenta, phosphomolybdate, and aniline blue in that order (GP1032, Servicebio Technology, Co., Ltd., Wuhan, China). PAS staining was performed with Periodic Acid–Schiff Staining Kit (GP1039, Servicebio Technology, Co., Ltd., Wuhan, China). Individual staining was carried out according to the manufacturer’s instructions.

### 4.6. 4D-Label-Free Proteomic Analysis

#### 4.6.1. Protein Extraction and Gel Separation

Liver tissue was taken and added to an appropriate amount of SDT (4% SDS, 150 mM Tris/HCl pH 8.0, 1 mM dithiothreitol (DTT), and protease inhibitor) lysate. After ultrasonic lysis, a boiling water bath was used for 10 min. centrifugation at 14,000× *g* for 15 min. The supernatant was filtered using a 0.22 µm centrifuge tube (8160, Corning Spin-X, St. Louis, MO, USA), and the filtrate was collected. Protein quantification was performed using the BCA reagent. The proteins were separated using SDS-PAGE. Staining was performed with Kaumas Brilliant Blue.

#### 4.6.2. Filter Aided Sample Preparation (FASP) Enzymatic Digestion

Eighty micrograms of protein solution were taken from each sample, and DTT (43819-5G, Sigma, St Louis, MO, USA) was added to a final concentration of 100 mM. After a boiling water bath was used for 5 min, 200 μL of the UA buffer (8 M Urea, 150 mM Tris-HCL, pH 8.5) was added and mixed, the mixture was then, centrifuged at 12,500× *g* for 15 min, and the filtrate was discarded. One hundred microliters of iodoacetamide (IAA) buffer (I1149-5G, Sigma, St. Louis, MO, USA) + 100 μL UA buffer were washed twice. One hundred microliters of a 40 mM NH_4_HCO_3_ solution (A6141-25G, Sigma, St. Louis, MO, USA) was added, and the mixture was centrifuged at 12,500× *g* for 15 min. This was repeated twice. A 40 μL Trypsin buffer was shaken at 600 rpm for 1 min and left at 37 °C for 16–18 h. This was centrifuged 20 μL of a 40 mM NH_4_HCO_3_ solution was then added. and the filtrate was collected by centrifugation. Peptides were desalted using a C18 Cartridge (WAT023590, Waters, MA, USA), lyophilized, and then re-solubilized by adding 40 μL of a 0.1% formic acid solution for peptide quantification (OD280).

#### 4.6.3. NanoElute Chromatography

Samples were separated using the NanoElute system (Bruker, Bremen, Germany) at nanoliter flow rates. Buffer Solution A was an aqueous 0.1% formic acid solution (A117, Thermo Fisher Scientific, Waltham, MA, USA) and Solution B was an aqueous 0.1% formic acid acetonitrile solution (acetonitrile is 100%) (1000304008, Merck, Kenilworth, NJ, USA). A chromatographic column was equilibrated with 100% Liquid A. Samples were separated by an autosampler onto an analytical column (IonOpticks, Fitzroy, Australia, 25 cm × 75 μm, C18 packing 1.6 μm) at a flow rate of 300 nL/min. The 1.5 h liquid phase gradient was as follows: at 0 min–3 min, liquid B 3%; at 3 min–73 min, liquid B linear gradient from 3% to 28%; at 73 min–80 min, liquid B linear gradient from 28% to 38%; at 80 min–85 min, liquid B linear gradient from 38% to 100%; at 85 min–90 min, liquid B maintained at 100%.

### 4.7. Mass Spectrometry Identification

The samples were separated by chromatography and analyzed by mass spectrometry in PASEF mode on a timsTOF Pro mass spectrometer (Bruker, Bremen, Germany). Analysis time was 90 min, detection was positive ion, the parent ion scan range was 100–1700 *m*/*z*, the ion flow rate 1/K0 was 0.6–1.6 V·s/cm^2^, the ion accumulation or release time was 100 ms, the ion utilization rate was 100%, the capillary voltage was 1500 V, the drying gas rate was 3 L/min, and then drying temperature was 180 °C. The PASEF settings were as follows: 10 MS/MS scans (the total cycle time was 1.16 s), a charge range of 0–5, a dynamic exclusion time of 0.4 min, an ion target intensity of 20,000, an ion intensity threshold of 2500, and a CID fragmentation energy of 42 eV.

### 4.8. Bioinformatics Analysis of Differentially Expressed Proteins

GO enrichment analysis of biological processes and molecular functions based on the DAVID bioinformatics database (https://david.ncifcrf.gov/gene2gene.jsp) (accessed on 21 February 2022) (Vision 2021). The KEGG Pathway database (http://www.genome.jp/kegg/pathway.html) (accessed on 28 January 2022) was used to analyze representative pathways for differentially expressed proteins and *p* ≤ 0.05 was used to represent the level of significance. Protein interaction analysis was performed using the String Database (https://string-db.org/) (accessed on 3 December 2021). The graphic visualization was created using Cytoscape (University of California, San Diego, CA, USA) (version 3.6.0).

### 4.9. Parallel Reaction Monitoring (PRM)

#### 4.9.1. Extraction of Peptides

The supernatant was removed by centrifugation after lysing the sample (1.5% SDS/100 mM Tris-Cl). The proteins in the solution were precipitated by acetone precipitation. After protein precipitation, 8 M Urea/100 mM Tris-Cl was added to the solution and then incubated with DTT (43819-5G, Sigma, St. Louis, MO, USA) for 1 h at 37 °C; subsequently, an alkylation reaction was carried out to seal the sulfhydryl groups. Protein concentrations were determined using the Bradford method (5000201, Bio-Rad, CA, USA). Digestion was carried out at a mass ratio of 1:50 (enzyme to protein). After digestion, the proteins were desalted, dried, and frozen at −20 °C for storage.

#### 4.9.2. Mass Spectrometry Detection

Mass spectrometry data were acquired using a Q Exactive HF mass spectrometer (Thermo Fisher Scientific, Waltham, MA, USA) in tandem with an UltiMate 3000 RSLCnano liquid phase liquid chromatography system (Thermo Fisher Scientific, Waltham, MA, USA). Peptide samples were dissolved in a loading buffer, aspirated by an autosampler, and separated on an analytical column (75 μm × 25 cm, C18, 1.9 μm, 120 Å). A liquid phase gradient was established using two mobile phases (Mobile phase A: 0.1% formic acid and 3% DMSO, and Mobile phase B: 0.1% formic acid, 3% DMSO and 80% ACN) to analyze the gradient. The flow rate of the liquid phase was set to 300 nL/min. The mass spectra were acquired in DDA mode, with each scan cycle containing one full MS scan (R = 60 K, AGC = 3 × 10^6^, max IT = 25 ms, scan range = 350–1500 *m*/*z*) and 20 subsequent MS/MS scans (R = 15 K, AGC = 1 × 10^5^, max IT = 50 ms). the higher-energy collisional dissociation (HCD) collision energy was set to 27. The screening window for the quadrupole was set to 1.4 Da. The dynamic exclusion time for ion repeat acquisition was set to 24 s.

#### 4.9.3. Mass Spectrometry Detection

The mass spectrometry data was searched in a database using Maxquant software. PRM method building and protein quantification were performed using Skyline software (https://skyline.ms/) (accessed on 16 June 2022) to obtain quantitative information on the target proteins and peptides.

### 4.10. Statistical Analysis

All experimental data obtained from mice are expressed as mean ± standard deviation (SD). The significance of differences between treatment groups was determined using Student’s *t*-test with GraphPad Prism 6.0 software (Dotmatics, San Diego, CA, USA). A level of *p* ≤ 0.05 was considered to indicate a statistically significant difference.

## Figures and Tables

**Figure 1 ijms-23-10045-f001:**
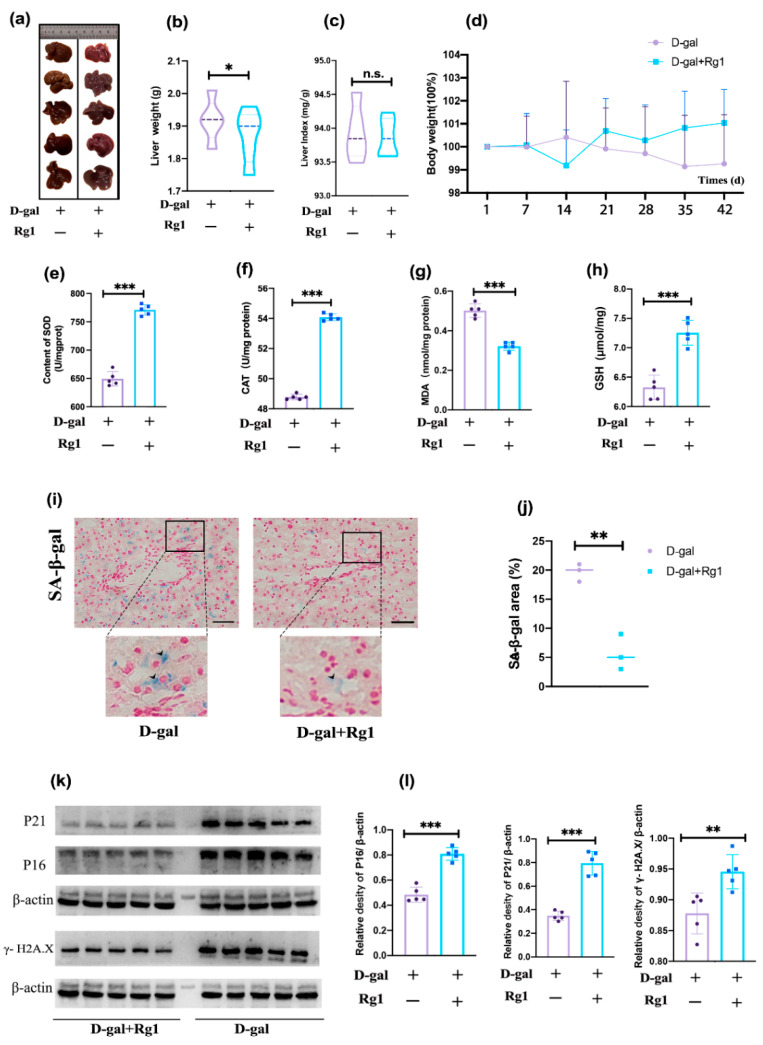
After D-gal treatment, the hepatocytes showed a decrease in the ability of anti-oxidative stress and senescence-associated cellular dysfunction, which can be alleviated by Rg1. (**a**) The appearance of the livers of each group. (**b**) The weight of livers was examined (*n* = 5/group). (**c**) The liver Index (liver weight/body weight) was measured (*n* = 5/group). (**d**) Daily weight was recorded to plot the weight gain curve. The levels of (**e**) SOD (**f**) CAT (**g**) MDA and (**h**) GSH in the liver of mice was examined (*n* = 5/group). (**i**) SA-β-gal staining showed the SA-β-gal+ cells and (**j**) the percentage of positive cells was detected (magnification, ×400, scale bar, 20 µm, *n* = 5/grp). Images were analyzed using Image J software (Version 1.52). (**k**) Western blotting for P16, P21, and γ- H2A.X expression levels. (**l**) The relative density of the bands was calculated. β-actin served as the internal control. Data represented by means ± SD in (**b**,**c**,**e**–**j**). (* *p* < 0.05; ** *p* < 0.01; *** *p* < 0.001 and n.s., not statistically significant. Student’s *t*-test).

**Figure 2 ijms-23-10045-f002:**
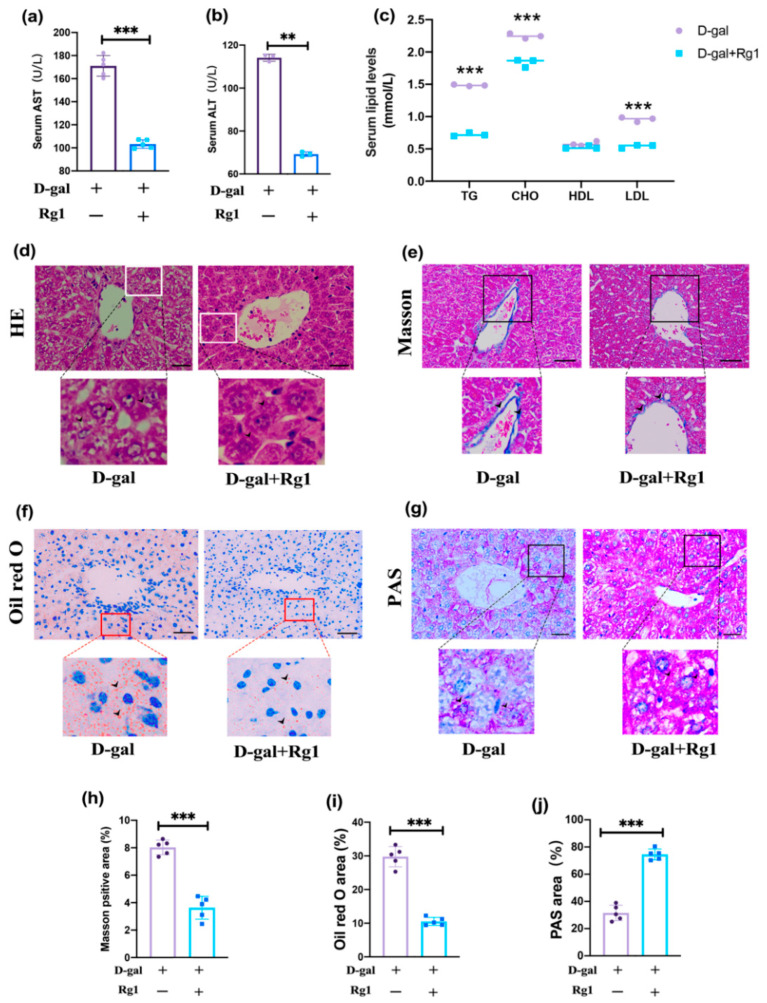
Rg1 alleviated the oxidative stress damage-related hepatocyte dysfunction: (**a**) The AST and (**b**) ALT levels in serum were detected (*n* = 5/group). (**c**) The serum lipid levels were measured (*n* = 5/group) (**d**) Hematoxylin and eosin staining (magnification, ×400, scale bar, 20 µm) (**e**,**h**) Masson staining and quantitative results of the percentage of cells in the liver of different experimental groups (magnification, ×400, scale bar, 20 µm). (**f**,**i**) Oil O Red staining was measured and quantitative analysis of lipid drops was detected (magnification, ×400, scale bar, 20 µm). (**g**,**j**) Periodic acid–Schiff staining and a quantitative analysis of hepatic glycogen were conducted (magnification, × 400, scale bar, 20 µm). Images were analyzed using Image J software (*n* = 5/group). Data represented by means ± SD in (**a**–**c**,**h**–**j**). (** *p* < 0.01 and *** *p* < 0.001, Student’s *t*-test).

**Figure 3 ijms-23-10045-f003:**
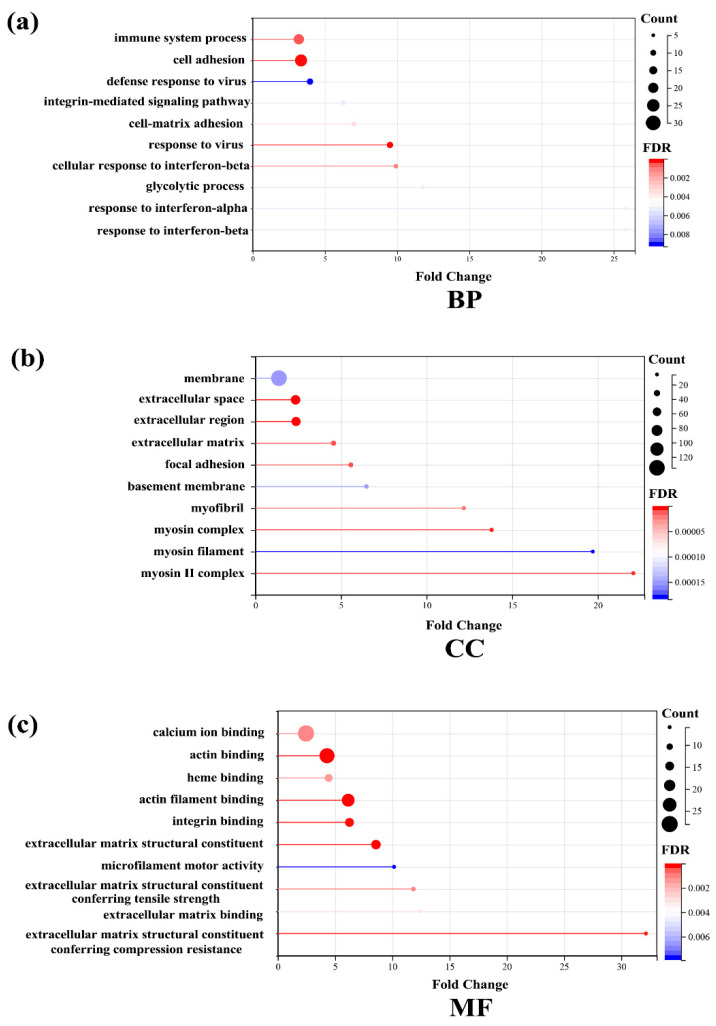
The top 10 gene ontology (GO) analyses of liver proteins. Proteins were classified according to (**a**) the cellular component (CC), (**b**) the biological process (BP) and (**c**) the molecular function (MF) by DAVID (2021 version).

**Figure 4 ijms-23-10045-f004:**
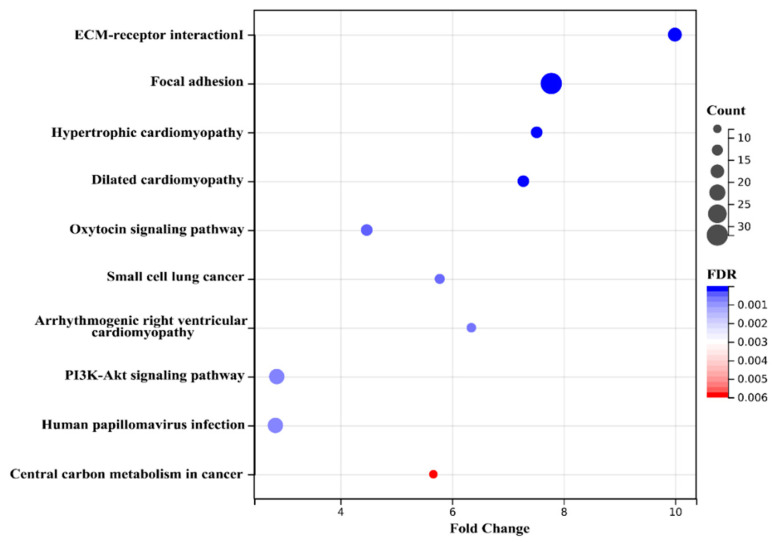
Top 10 enriched KEGG pathways of proteins in the liver. The false discovery rate (FDR) (lower and more intense rates in blue) represents the enriched degree. The size of the dots represents the number of proteins in this pathway.

**Figure 5 ijms-23-10045-f005:**
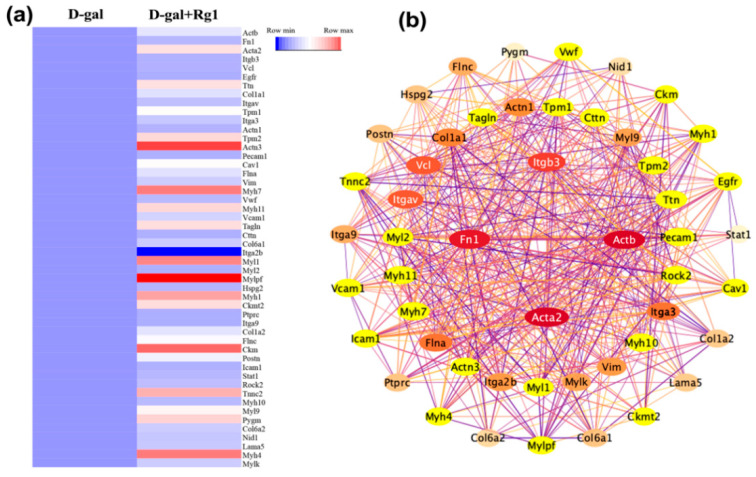
The expression abundance and protein interaction network of key proteins: (**a**) the heat map of the key proteins; (**b**) PPI of key proteins.

**Figure 6 ijms-23-10045-f006:**
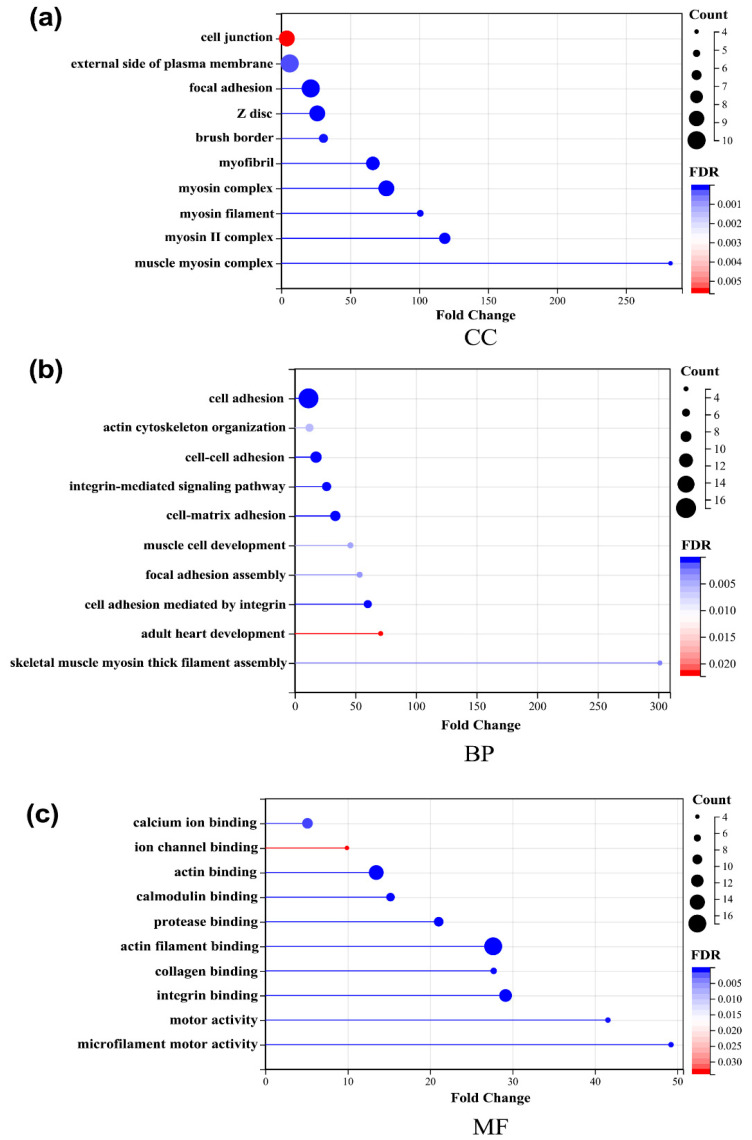
Gene ontology (GO) analysis of key proteins. Proteins were classified according to (**a**) the cellular component (CC), (**b**) the biological process (BP) and (**c**) the molecular function (MF) by DAVID (version 2021).

**Figure 7 ijms-23-10045-f007:**
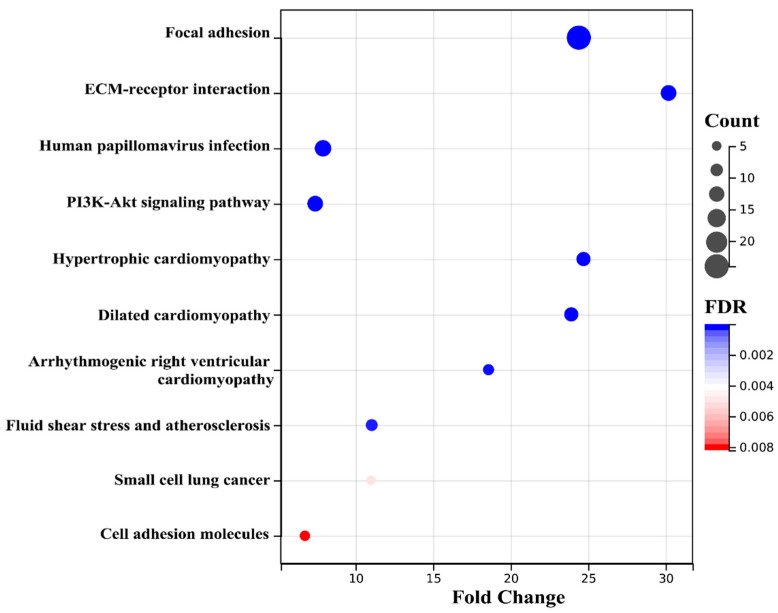
Top 10 enriched KEGG pathways of key proteins. The false discovery rate (FDR) (lower more rates intense in blue) represents the enriched degree. The size of the dots represents the number of proteins in this pathway.

**Figure 8 ijms-23-10045-f008:**
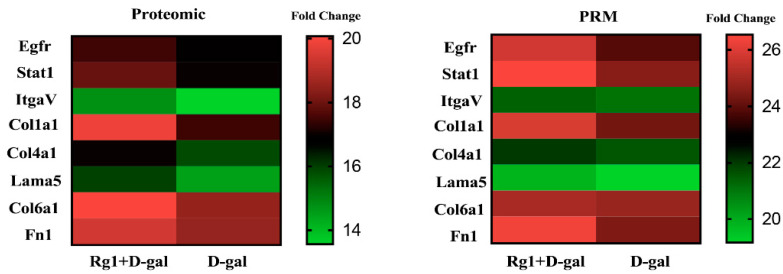
The relative fold change of the eight proteins in D-gal + Rg1: D-gal by 4D-label free proteomics and PRM. (each sample consists of the liver tissue of 5 mice).

## Data Availability

Not applicable.

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
