# Peer review of "Revealing the Therapeutic Targets and Mechanism of Ginsenoside Rg1 for Liver Damage Related to Anti-Oxidative Stress Using Proteomic Analysis"

_ijms, 2022, doi:10.3390/ijms231710045_

Round 1
Reviewer 1 Report
The paper provides valuable information about the physiological and signaling effects of ginsenoside Rg1. The data significantly add to the current knowledge on ginsen root derived drugs and their application in medicine.
The paper needs language editing (e.g. in the Abstract: "Proteomic analysis suggested that Rg1 may regulate hepatocyte metabolism through ECM-23 Receptor, PI3K-AKT pathway . And EGFR, STAT1 may be the key protein." ?)
Author Response
Response to Reviewer 1 Comments
Point 1: The paper needs language editing (e.g. in the Abstract: "Proteomic analysis suggested that Rg1 may regulate hepatocyte metabolism through ECM-23 Receptor, PI3K-AKT pathway . And EGFR, STAT1 may be the key protein." ?)
Response 1: Thank you very much for your suggestions on our manuscript. We have contacted MDPI's English editing service and revised the English language of this manuscript. Please find attached the certificate of the English editor.

Reviewer 2 Report
I consider that it is a relevant work, but I think it could be improved.
General comments
Line 94
(a) The appearance of the livers of each group
Nothing is described in this sentence if the appearance is different, so include some description. Whether or not there were differences and what they were. (texture, color)
Line 178
2.4. Bioinformatics analysis of key proteins
In this section, nothing is said about the three categories mentioned above: biological processes (BP), molecular functions (MF), and cellular components (CC). Which of them contributes more, or how are these processes linked?
Material and methods
This section must include the value of the dose of Rg1 used, in addition to containing the reference where this dose had already been used previously. And not only the reference since the reader is left without this information if he cannot access the reference.
Line 300 4.1. Animal experiments
Rg1 was purchased from Biological Reagents (MCE, Cat. No. HY-N0045 or Jilin Hongjiu Biotechnology Co., Ltd.) at the doses already mentioned in past publications [38].
When including females in the studies, the authors should inform in what stage of the cycle the animals were in and if they regulated their hormonal cycles.
Line 302
C57BL/6J mice (6-8 weeks, equal numbers of male and female mice)
In this section, the authors must describe why they used this strain of mice and what was the advantage over other mice at the biochemical or physiological level that makes them ideal for this model because none of this is included in the introduction or the material and methods section.
C57BL/6J mice
Authors should include the dose of pentobarbital used to anesthetize the mice.
Line 312
Mice in the modelling and dosing group were put to death at the end of modelling using CO2 and deep anaesthesia with pentobarbital.
Line 314
4.2. Antioxidant enzymes and redox products
To determine the concentrations of proteins and enzymes, some chemical reactions are carried out. So, after the brief description, it is necessary to include a method reference. And it should not only be said that it was made according to the kit manufacturer's instructions.
catalase (CAT) (ref), glutathione peroxidase (GSH) (ref), superoxide dismutase (SOD) (ref) and Malondialdehyde (MDA)(ref) kits according to the instructions.
Line 320
4.3. Senescence associated
β-galactosidase (SA-β-gal) staining (ref) and Oil Red O staining (ref)
Line 326
4.4. Immunoblot assay
Tissue proteins were extracted after ultrasonic lysis, and the protein concentration was measured by BCA (ref). Electrophoresis was carried out using SDS-page gels (ref)
Line 336
4.5 Paraffin section
All stains
hematoxylin-eosin staining (HE) (ref)
Masson staining (ref)
PAS staining (ref)
Line 350
Protein quantification was performed using the BCA (ref)
Line 353
4.6.2. FASP enzymatic digestion
Include the meaning of the abbreviation before
DTT The meaning of this abbreviation is included after.
Line 395
incubated with dithiothreitol (DTT)
UA buffer
Write the numbers in the formula in subscripts.
NH4HCO3
Line 410
The meaning of the abbreviation HCD is:
Higher-energy collisional dissociation (HCD)
HCD collision energy was
Figure 1, part h
The authors want to show the area that is then enlarged. But this would look better with fewer stripes. Instead of a continuous line, one joins the small square with the big square. It could be done with dotted lines -------- it would look better- - - -
This same recommendation for figure 2 in the stains (d), (e), (f), and (g).
(d) the wrong choice of red; is lost on staining. choose another color
(e) Good choice of black color.
(f) Good choice of red color.
(g) Good choice of black color, but thicken the line more, make it like (e)
But in all of them, degrade the line that connects the two boxes, either with a dotted line ------ or with a smaller thickness. This will allow you to highlight the area of tissue you want to show.
Figure 5 Line 199
(b) PPI of key proteins. An explanation of the meaning of the protein colors should be included. To understand the levels, they represent.
I do not know if the program is the one that gives the colors. But I suggest if it is possible to change the red color that is in the background because the black letter makes it difficult to read what it says.
Or leave the background red but change the black font to white or light.
Discussion
Line 282
Here it will be important to clarify the concentration of Rg1 as an antioxidant since some antioxidants have a prooxidant action at high concentrations
However, the EGFR signaling pathway is a double-edged sword. Although the EGFR signaling system has been shown in many studies to play a central role in the repair and regenerative response to liver injury and inflammation, including inhibition of intrahepatic lipid accumulation [35].
Author Response
Thank you very much for your sincere and detailed suggestions on our article. We have made the following changes in response to your suggestions.
We have marked the changes in red in the manuscript. Deletions and parts that require special clarification are additionally marked. The location of the corresponding changes to the different issues is indicated in the response letter.
If the online responses are difficult to read, please see the attachment. We have also uploaded a word version of the response letter.
Thank you again for your careful guidance, from which we have benefited greatly.
Point 1: 1.(a) The appearance of the livers of each group
Nothing is described in this sentence if the appearance is different, so include some description. Whether or not there were differences and what they were. (texture, color)
Response 1: We have added a description of the morphology and texture of the liver in Line 78:
The liver appearance of the D-gal group was yellow-tinged and greasy while the livers of Rg1+D-gal group were pink-red(Figure 1 a)
Point 2: Bioinformatics analysis of key proteins
In this section, nothing is said about the three categories mentioned above: biological processes (BP), molecular functions (MF), and cellular components (CC). Which of them contributes more, or how are these processes linked?
Response 2: Gene ontology ( Gene ontology , GO ) is a method and process to systematically annotate the properties of species genes and their products. In fact, ontology refers to the method of elaborating what is known, describing what is observable and how it is related to each other. GO was created primarily to address the difficulties of communication and information sharing caused by differences in terminology between different fields in biology. At the same time, GO is a machine-readable language to facilitate large-scale biological and genetic analysis. In order to achieve such a goal, each entry (GO term) in the gene annotation usually contains an item name, a unique ID, a named domain to indicate the domain to which it belongs and definition of a reference source。At the same time,the annotation of gene ontology is mainly based on the following three perspectives:
1. Cellular component(CC); each part of the cell and the extra cellular environment of the cell.
2.Molecular function(MF); the main activity of a gene product at the molecular level, such as binding and enzymatic catalysis .
3.Biological process(BP); a process or collection of molecular events, events or actions that can define beginning and end, that occurs in integrated living units, such as cells, tissues, organs, and organisms.
Here, we use GO analysis in order to see in which cellular functional regions the significantly differentially expressed proteins are mainly concentrated. Therefore, there should be a parallel relationship between the three entries. For reasons of parallelism, each part is relatively independent of the other, so no additional explanation is given in the manuscript.
Point 3: Material and methods
This section must include the value of the dose of Rg1 used, in addition to containing the reference where this dose had already been used previously. And not only the reference since the reader is left without this information if he cannot access the reference.
Animal experiments
Rg1 was purchased from Biological Reagents (MCE, Cat. No. HY-N0045 or Jilin Hongjiu Biotechnology Co., Ltd.) at the doses already mentioned in past publications [38].
Response 3: Thank you for your suggestion, we have indicated the exact dosage of Rg1 and have cited additional open access references.[Line 328-330]:
Point 4: When including females in the studies, the authors should inform in what stage of the cycle the animals were in and if they regulated their hormonal cycles.
Response 4:Thank you for your suggestion. Over the last few years we have observed the effect of Rg1 in several tissues in order to discuss whether Rg1 causes adverse effects. Such as [ PMID: 27294914] [PMID: 34364981][PMID: 31885611][PMID: 26730750]
The results show that a safe dose (20mg/kg. day) of Rg1 does not cause tissue damage and also provides good relief of ROS accumulation in a variety of organs.
Among them, for the reproductive system, our group has published a previous article [PMID: 28178855], [PMID: 33179093] and [PMID: 34463232]. These experiments observed changes in ovarian, uterine and kinetic cycles in mice with D-gal modelling-induced oxidative stress injury. The results showed that mice in both the Rg1 and Rg1+D-gal groups preserved a normal estrous cycle during the experimental cycle (compared with control group). Therefore, we conclude that 20 mg/kg of Rg1 does not affect the estrous cycle in female mice.
We have added the following at [Line 324].
Point 5: C57BL/6J mice (6-8 weeks, equal numbers of male and female mice)
In this section, the authors must describe why they used this strain of mice and what was the advantage over other mice at the biochemical or physiological level that makes them ideal for this model because none of this is included in the introduction or the material and methods section.
Response 5: Thank you for your suggestion. We chose C57BL/6J based on its stable genetic background as an inbred mouse. Another reason is that C57BL/6J has finished its whole genome sequenced[PMID: 12466850]. Therefore it is a good choice as a subject for proteomics analysis. We has added this information in the Line 320 of manuscript.
Point 6: Authors should include the dose of pentobarbital used to anesthetize the mice.
Mice in the modelling and dosing group were put to death at the end of modelling using CO2 and deep anaesthesia with pentobarbital.
Response 6: Thank you for your suggestion, we have synthesized papers [PMID: 6473905] and finally used 50 mg/kg of pentobarbital for anesthetic sedation followed by death by asphyxia using high CO2 concentrations. The concentrations are described in the Materials and methods section with the addition of: [Line 341].
Also, errors in the use of CO2 have been fixed. [Line 340]
Point 7: Antioxidant enzymes and redox products
To determine the concentrations of proteins and enzymes, some chemical reactions are carried out. So, after the brief description, it is necessary to include a method reference. And it should not only be said that it was made according to the kit manufacturer's instructions.
catalase (CAT) (ref), glutathione peroxidase (GSH) (ref), superoxide dismutase (SOD) (ref) and Malondialdehyde (MDA)(ref) kits according to the instructions.
Senescence associated
β-galactosidase (SA-β-gal) staining (ref) and Oil Red O staining (ref)
Immunoblot assay
Tissue proteins were extracted after ultrasonic lysis, and the protein concentration was measured by BCA (ref). Electrophoresis was carried out using SDS-page gels (ref)
Paraffin section
All stains
hematoxylin-eosin staining (HE) (ref)
Masson staining (ref)
PAS staining (ref)
Protein quantification was performed using the BCA (ref)
Response 7: Thank you very much for your suggestions. We have made changes in Lines 343-368, Lines 372-373, Lines 376-398.
The revision adds a brief description of the experimental method, the company that produces the reagent and quotes its stock number in the relevant company so that other researchers can quickly find the product and its instructions. Thank you very much for your patient and detailed guidance on our articles.
Point 8: FASP enzymatic digestion
Include the meaning of the abbreviation before
DTT The meaning of this abbreviation is included after.
incubated with dithiothreitol (DTT)
UA buffer
Write the numbers in the formula in subscripts.
NH4HCO3
The meaning of the abbreviation HCD is:
Higher-energy collisional dissociation (HCD)
HCD collision energy was
Response 8: We have supplemented the list of abbreviations and checked where the abbreviations first appear, and the reagents in this section have been supplemented with explanations.
[Line 402-405], [Line 410-421] and [Line 424-436]
Also, a list of abbreviations is provided at the end of the manuscript.[Line 514]
Point 9:Figure 1, part h
The authors want to show the area that is then enlarged. But this would look better with fewer stripes. Instead of a continuous line, one joins the small square with the big square. It could be done with dotted lines -------- it would look better- - - -
This same recommendation for figure 2 in the stains (d), (e), (f), and (g).
(d) the wrong choice of red; is lost on staining. choose another color
(e) Good choice of black color.
(f) Good choice of red color.
(g) Good choice of black color, but thicken the line more, make it like (e)
But in all of them, degrade the line that connects the two boxes, either with a dotted line ------ or with a smaller thickness. This will allow you to highlight the area of tissue you want to show.
Response 9: Thank you for your nice guidance. We have modified the solid lines in Figure 1 (i) and Figure 2 (d-g) to dashed lines. We have also modified the color of the border in Figure 2 (d).
Point 10: PPI of key proteins. An explanation of the meaning of the protein colors should be included. To understand the levels, they represent.
I do not know if the program is the one that gives the colors. But I suggest if it is possible to change the red color that is in the background because the black letter makes it difficult to read what it says.
Or leave the background red but change the black font to white or light.
Response 10: We have marked the font on the red background in the image in white and have explained what the different colors in the PPI represent at the results section in the manuscript.[Line 196-199].
Point 11: Discussion
Here it will be important to clarify the concentration of Rg1 as an antioxidant since some antioxidants have a prooxidant action at high concentrations
However, the EGFR signaling pathway is a double-edged sword. Although the EGFR signaling system has been shown in many studies to play a central role in the repair and regenerative response to liver injury and inflammation, including inhibition of intrahepatic lipid accumulation.
Response 11: Thank you again for your advice. We have added the appropriate Rg1 dose to the Materials and Methods. [Line 328]

Reviewer 3 Report
The ginsenoside Rg1is an active component of ginseng, an ancient traditional medical herb. Extensive studies have been performed on the pharmacological effects of Rg1. In the present study, Hou et al. has examined the effects of Rg1 on the D-galactose-induced hepatic oxidative stress damage model mice. Methods used are biochemical analyses of the liver and serum, proteomics of the liver protein expression, and bioinformatics. Overall, the study is well-designed and carried out. However, there are a number of major and minor flaws in the manuscript in its present form, as described below.
Major points:
1) In Figure 1 and 2, there are only two experimental groups: D-gal+/Rg1- and D-gal+/Rg1+. How about the negative control group: D-gal-/Rg1-? In fact, the same research group (led by Yaping Wang) has recently published a related paper “Ginsenoside Rg1 attenuates premature ovarian failure of D-gal induced POR mice…” in Endocrine, Metabolic & Immune Disorders – Drug Targets 2022, 22, 318-337. In this article, the authors (Xio-Hu Liu et al.) included a negative control. In the present manuscript by Hou et al., the authors should at least explain why the negative control was omitted.
2) The style of abstract and references are not consistent with those of the journal IJMS. The references require extensive editing. The journal also requires a list of abbreviations. The abbreviations should accompany full names at the first appearance.
Minor points:
3) There are some points where the usage of English is not appropriate: Examples are line 15 (unknow) and line 276 (to alter).
4) Figure 1 and 2: If P values are <0.01 and <0.001, they should be labelled as ** and ***, respectively.
5) Line 216 (Table 3): Is it Table S3? There is no Table 3 in the main text.
6) Labelling in Figure 3 and 4 are too small to be readable.
Author Response
Thank you very much for your sincere and detailed suggestions on our article. We have made the following changes in response to your suggestions.
We have marked the changes in red in the manuscript. Deletions and parts that require special clarification are additionally marked. The location of the corresponding changes to the different issues is indicated in the response letter.
If the online version is difficult to read, please see the attachment. We have also uploaded a Word version of the response letter.
Thank you again for your careful guidance, from which we have benefited greatly.
Major points:
Point 1: In Figure 1 and 2, there are only two experimental groups: D-gal+/Rg1- and D-gal+/Rg1+. How about the negative control group: D-gal-/Rg1-? In fact, the same research group (led by Yaping Wang) has recently published a related paper “Ginsenoside Rg1 attenuates premature ovarian failure of D-gal induced POR mice…” in Endocrine, Metabolic & Immune Disorders – Drug Targets 2022, 22, 318-337. In this article, the authors (Xio-Hu Liu et al.) included a negative control. In the present manuscript by Hou et al., the authors should at least explain why the negative control was omitted.
Response 1: Thank you very much for your sincere and constructive questions.
Negative control is very important for controlling the effect of confounding factors on experiments. In the article on Rg1 for D-gal-induced ovarian damage[PMID: 34463232], the PBS group, the Rg1 group and the D-gal group were used for experimental comparison. To illustrate the effect of Rg1 effect and the changes of the D-gal group. In the current experiment, however, the D-gal group was used directly as a control group. The differences in experimental design between these experiments are based on the following considerations:
- We have published two previous studies on the role of Rg1 in oxidative stress damage to the liver[1, 2]. Both experiments were conducted using the negative control group, the D-gal group and the D-gal+Rg1 group to verify the protective effect of Rg1 on the liver. In addition, we have also tested the anti-oxidative stress effect of Rg1 in different tissues in past experiments[3-7]. These experiments all used D-gal as a way of modelling the induction of oxidative stress. The results of these experiments show that Rg1 does not cause cytotoxic effects and damage to tissues at a safe doses (20mg/kg. day). Therefore, we extended the dose of these experiments and concluded that Rg1 treatment should be safe. On the other hand, based on the principle of using the least number of animals possible in the ethics of animal experimentation, we thought that the D-gal modelling group could be used directly as a control group on the basis of the preliminary experiments.
- Globally, D-gal has been used in many experiments as a drug for accelerated aging models due to its induction of oxidative stress damage[8]. In particular, it is widely used in liver research as a drug to mimic oxidative stress injury[9] which is a stable and mature way of modelling. On the other hand, senescence due to oxidative stress damage has a specific phenotype and changes. SA-β-gal staining and P16 protein are considered to be the biomarkers of senescence[10, 11]. P21, also known as cyclin-dependent kinase inhibitor 1, is a protein that is closely associated with DNA damage and cell cycle arrest[10]. The expression of these markers is very low in the normal cells and is significantly increased when cellular senescence leads to cell cycle arrest. Therefore, basing on the low expression of these markers in normal cells. We conclude that when D-gal results in abnormally high P21, P16 and a significant increase in SA-β-gal positivity in tissues, it can be considered as successfully inducing oxidative stress-related senescence model.
- Finally, using the disease model as a control group may be a viable modus operandi in drug target finding experiments. Such as the experiment to study the improvement of fatty liver in mice on a ketogenic diet by probiotics, simulated toddler feeding was used as a control group, while a negative control group was not used[12]. The experiment with Ocimum x africanum Essential Oil, gastric cancer cell lines were used as controls for the analysis[13].
Therefore, based on the basis of past experiments, in this experiment we considered the D-gal modelling group as the control group. And Rg1 was considered as the treatment group. Both molecular biology and proteomics analyses were performed based on the above groupings.
Thank you very much for your suggestion and we have explained the reasons for the grouping in the Materials and Methods section of the manuscript. [Line 328-333]
References of Response 1:
- Qi, R., et al., Ginsenoside Rg1 protects against d -galactose induced fatty liver disease in a mouse model via FOXO1 transcriptional factor. Life Sciences, 2020. 254.
- Ming-He, et al., Ginsenoside Rg1 attenuates liver injury induced by D-galactose in mice. Experimental & Therapeutic Medicine, 2018.
- Jing, L., et al., Protective Effect of Ginsenoside Rg1 on Hematopoietic Stem/Progenitor Cells through Attenuating Oxidative Stress and the Wnt/β-Catenin Signaling Pathway in a Mouse Model of d-Galactose-induced Aging. International Journal of Molecular Sciences, 2016. 17(6).
- Wang, Z., et al., Ginsenoside Rg1 prevents bone marrow mesenchymal stem cell senescence via NRF2 and PI3K/Akt signaling. Free Radical Biology and Medicine, 2021(16).
- He, L., et al., Ginsenoside Rg1 improves fertility and reduces ovarian pathological damages in premature ovarian failure model of mice. Experimental Biology & Medicine, 2017. 242(7): p. 1535370217693323.
- Yue, et al., Effects of Ginsenoside Rg1 Regulating Wnt/β-Catenin Signaling on Neural Stem Cells to Delay Brain Senescence. Stem cells international, 2019. 2019: p. 5010184-5010184.
- Yanling, et al., Mechanism of ginsenoside Rg1 renal protection in a mouse model of D-galactose-induced subacute damage.
- Azman, K.F. and R. Zakaria, d -Galactose-induced accelerated aging model: an overview. Biogerontology, 2019. 20(8).
- Azman, K.F., A. Safdar, and R. Zakaria, D-galactose-induced liver aging model: Its underlying mechanisms and potential therapeutic interventions. Experimental Gerontology, 2021. 150(10): p. 111372.
- Mchugh, D. and J. Gil, Senescence and aging: Causes, consequences, and therapeutic avenues. The Journal of Cell Biology, 2018(1).
- Alcorta, D.A., et al., Involvement of the cyclin-dependent kinase inhibitor p16 (INK4a) in replicative senescence of normal humanfibroblasts. Proceedings of the National Academy of Sciences of the United States of America, 1996. 93(24): p. 13742-13747.
- Cm, A., et al., Probiotics counteract hepatic steatosis caused by ketogenic diet and upregulate AMPK signaling in a model of infantile epilepsy. 2022.
- Boonyanugomol, W., et al., Endoplasmic Reticulum Stress and Impairment of Ribosome Biogenesis Mediate the Apoptosis Induced by Ocimum x africanum Essential Oil in a Human Gastric Cancer Cell Line. Medicina, 2022. 58(6): p. 799.
Point 2: The style of abstract and references are not consistent with those of the journal IJMS. The references require extensive editing. The journal also requires a list of abbreviations. The abbreviations should accompany full names at the first appearance.
Response 2: Thank you for your suggestion. We have revised the formatting of the abstract and references. We have also checked the abbreviations in the manuscript to ensure that they are accompanied by an explanation when they first appear. A list of abbreviations is included at the end of the manuscript.
Minor points:
Point 3: There are some points where the usage of English is not appropriate: Examples are line 15 (unknow) and line 276 (to alter).
Response 3: Thanks to your comments. We have changed the language errors and had the manuscript checked by a native English speaking colleague.
Point 4: Figure 1 and 2: If P values are <0.01 and <0.001, they should be labelled as ** and ***, respectively.
Response 4: Thanks to your comments. We have re-labelled the p-values less than 0.01 and 0.001 in Figure 1 and Figure 2.
Point 5: Line 216 (Table 3): Is it Table S3? There is no Table 3 in the main text.
Response 5: We are very sorry for this misquote due to an oversight on our part. We have corrected it.
Point 6: Labelling in Figure 3 and 4 are too small to be readable.
Response 6: Thanks to your comments. We have enlarged the font size in Figure 3 -4 and Figure 6-7.

Round 2
Reviewer 3 Report
The authors have adequately addressed all of the comments by Reviewer. However, the style of References still does not follow that by adopted by IJMS.